# Dynamic Modelling of Phosphorolytic Cleavage Catalyzed by Pyrimidine-Nucleoside Phosphorylase

**Robert T. Giessmann** [1,*,†] **, Niels Krausch** [1,†] **, Felix Kaspar** [1] **,**
**Mariano Nicolas Cruz Bournazou** [2,3] **, Anke Wagner** [1,4] **, Peter Neubauer** [1]
**and Matthias Gimpel** [1]

[1]   Laboratory of Bioprocess Engineering, Department of Biotechnology, Technische Universität Berlin, Ackerstr. 76, ACK24, D-13355 Berlin, Germany; n.krausch@tu-berlin.de (N.K.); f.kaspar@tu-braunschweig.de (F.K.); anke.wagner@tu-berlin.de (A.W.); peter.neubauer@tu-berlin.de (P.N.); matthias.gimpel@tu-berlin.de (M.G.)

[2]   Institute of Chemical and Bioengineering, Department of Chemistry and Applied Biosciences, ETH Zürich, Vladimir-Prelog-Weg 1, 8093 Zurich, Switzerland; n.cruz@datahow.ch

[3]   DataHow AG, Vladimir-Prelog-Weg 1, 8093 Zurich, Switzerland

[4]   BioNukleo GmbH, Ackerst. 76, D-13355 Berlin, Germany

*   Correspondence: r.giessmann@tu-berlin.de

†   Authors contributed equally to this paper.

**Abstract:** Pyrimidine-nucleoside phosphorylases (Py-NPases) have a significant potential to contribute to the economic and ecological production of modified nucleosides. These can be produced via pentose-1-phosphates, an interesting but mostly labile and expensive precursor. Thus far, no dynamic model exists for the production process of pentose-1-phosphates, which involves the equilibrium state of the Py-NPase catalyzed reversible reaction. Previously developed enzymological models are based on the understanding of the structural principles of the enzyme and focus on the description of initial rates only. The model generation is further complicated, as Py-NPases accept two substrates which they convert to two products. To create a well-balanced model from accurate experimental data, we utilized an improved high-throughput spectroscopic assay to monitor reactions over the whole time course until equilibrium was reached. We examined the conversion of deoxythymidine and phosphate to deoxyribose-1-phosphate and thymine by a thermophilic Py-NPase from *Geobacillus thermoglucosidasius*. The developed process model described the reactant concentrations in excellent agreement with the experimental data. Our model is built from ordinary differential equations and structured in such a way that integration with other models is possible in the future. These could be the kinetics of other enzymes for enzymatic cascade reactions or reactor descriptions to generate integrated process models.

**Keywords:** enzymatic reaction; reversible reaction; dynamic modelling; pyrimidine-nucleoside phosphorylase; spectroscopic assay; process kinetics; ODE model

## 1. Introduction

Pyrimidine-nucleoside phosphorylases (Py-NPases) are highly versatile enzymes used for the production of pharmaceutically relevant nucleoside derivatives and pentose-1-phosphates. Generally, nucleoside phosphorylases catalyze, in the presence of phosphate, the reversible conversion of a nucleoside to the corresponding pentose-1-phosphate and nucleobase (Figure 1). Due to the low yields of modified nucleosides or pentose-1-phosphates via conventional synthetic chemistry, nucleoside phosphorylases have become attractive tools in their biocatalytic preparation [1–3]. Recently, thermophilic Py-NPases have attracted increased interest, as they combine several favorable properties,

such as long shelf life due to their thermal stability, an excellent tolerance towards harsh reaction conditions, high turnover rates, and a broad substrate spectrum [4,5].

**Figure 1.** **Schematic and chemical illustration of an enzymatic nucleoside phosphorylation.** (**a**) Schematic drawing of the proposed mechanics for an enzymatic nucleoside phosphorylation reaction as basis for the generation of the differential–dynamical model. Enzyme (E), nucleoside (N), and phosphate (P) react in a three-particle collision towards the enzyme complex (EC), which decays without other intermediates into enzyme, pentose-1-phosphate (S1P), and free nucleobase (B). Both reactions can occur in the other direction, as well; (**b**) chemical structures of an enzymatic phosphorylation using the example of the enzyme pyrimidine-nucleoside phosphorylase (Py-NPase; E) catalyzed reaction of the nucleoside deoxythymidine (N) and ortho-phosphate (P) to the free nucleobase thymine (B) and deoxyribose-1-phosphate (S1P).

However, their industrial use is hampered by a lack of models which integrate the understanding of their behavior in enzymatic reactions over the full time course towards the reaction's dynamic equilibrium. Previous research has focused on either: (1) Integrated processes, mainly with transglycosylation and/or product removal reactions, which renders modelling of the complete process unfeasible because of its complexity; or (2) Michaelis–Menten conditions, i.e., reactions in which one of the substrates (typically phosphate) is present in excess over the other substrate, and only initial rates are measured (reviewed in [6]). This is because the Michaelis–Menten assumptions are only fulfilled in the quasi-linear range of conversion at the very start of the enzymatic reaction. Only in this time frame one can observe a constant conversion rate. Invariably, this only allows for the investigation of the dependence of the initial rate of the reaction on the concentration of a substrate and does not permit the evaluation of the whole time-course [6].

In industrial applications, the stoichiometric and quantitative conversion of substrates is highly anticipated. These requirements are only met when the reaction approaches its thermodynamic equilibrium, hence giving maximum product yield. Counteracting the accessibility of deoxyribose-1-phosphate is the fact that the equilibrium for nucleoside phosphorylation reactions is strongly in favor of the substrates ($K_{eq}$ = 0.03–0.10 for pyrimidines [7,8], and $K_{eq}$ = 0.01–0.02 for purines [9,10]). To increase the concentration of desired products, it is therefore necessary to push the equilibrium, e.g., by increasing the phosphate concentration. Despite the clear need for a Py-NPase model describing those industrially relevant conditions, there has been no report of a suitable model so far.

Models of ordinary differential equations (ODEs) derived from elementary reaction steps and from law of mass action kinetics ("differential–dynamical models") present an attractive solution to many biotechnological problems. Their modularity allows for the combination of models of different scales, such as the progression of an enzyme reaction with a substrate feeding profile. Differential–dynamical models have been used to describe, for example, enzymatic cellulose hydrolysis (reviewed in [11]), the production of enantiopure amines from a racemic mixture [12], the continuous production of lactobionic acid from lactose [13], or symmetric two-educts/one-product carboligations [14]. The rate laws of differential–dynamical models are usually derived from an underlying mechanical model. This enables chemical reaction engineering across different conditions and scales [15]. The ultimate promise of differential–dynamical models is the model-based design of dynamic experiments [16], which are

favorable for biotechnological applications [17] and allow the in silico predictability of economic production processes [18], even for processes where the experimental information is scarce [19].

In this work, we present experimental data deduced from the reaction monitoring of small-scale Py-NPase reactions via a UV/Vis spectroscopy-based assay. Subsequently, we report the development of a differential–dynamical model for the Py-NPase-mediated biocatalytic preparation of deoxyribose-1-phosphate from thymidine.

## 2. Materials and Methods

### 2.1. Materials

All chemicals used in this study were of analytical grade and used without further purification. The water used in all solutions was deionized to 18.2 MΩ·cm with a water purification system from Werner. Deoxythymidine was purchased from Carbosynth. Thymine and phosphate ($KH_2PO_4$) were purchased from Sigma–Aldrich. Tris (2-Amino-2-(hydroxymethyl)propane-1,3-diol) was of buffer grade and purchased from Carl Roth.

Tris buffer was prepared as a 50 mM solution, and the pH was adjusted to 9.0 using 1 M HCl. Phosphate was prepared as a 1 M stock solution in 50 mM Tris buffer, and the pH was subsequently adjusted to 9.0 using 1 M NaOH. Deoxythymidine, and thymine stock solutions were prepared in different concentrations (ranging from 1 to 10 mM) by adding 50 mM of Tris buffer (of pH 9.0; the final pH of the prepared solution was found to be 9.0 as well) and treated with ultrasound to facilitate full dissolution.

The enzyme under investigation was a Py-NPase (EC 2.4.2.2, NCBI sequence accession number WP_041270053.1) from *Geobacillus thermoglucosidasius* (DSM No.: 2542). After IPTG-induced recombinant overexpression, the N-terminally $His_6$-tagged Py-NPase was purified from *E. coli* BL21 using Ni-NTA affinity chromatography, as described previously [20]. Purity was determined by SDS-PAGE analysis and found to be >90%. Subsequently, the enzyme was dialyzed against 2 mM potassium phosphate buffer, pH 7.0 (measured at 25 °C), and stored until use at +4 °C at a concentration of 3.69 mg/mL, as judged by NanoDrop analysis (calculated with 0.48 absorption units (AU) at 280 nm = 1 mg/mL). One unit (1 U) of enzyme activity was defined as the conversion of 1 μmol of deoxythymidine per minute in a 1 mL assay mixture of 2 mM deoxythymidine and 75 mM phosphate in 50 mM Tris buffer at a reaction temperature of 40 °C and at pH = 9.0 (measured at 25 °C), as determined by the method described later. The molecular weight of the enzyme was 47.6 kDa, as calculated from its amino acid sequence. The used enzyme preparation had an activity of approximately 0.46 U/mg.

UV/Vis transparent 96-well plates (UV-STAR F-Bottom #655801, purchased from Greiner Bio-One) were used to host the solutions for UV/Vis spectroscopy.

### 2.2. Experimental

Phosphate and deoxythymidine concentrations were varied in the range of 2–80 mM and 0.8–5 mM, respectively, in the assay mixture. The final enzyme concentration in the assay mixture was in the range of 12.5–50 μg/mL. This corresponds to an enzyme monomer concentration of 0.26–1.05 μM, as calculated from its molecular weight.

Reaction mixtures were prepared in 1.5 mL microreaction tubes. Appropriate amounts of phosphate and deoxythymidine stock solutions were added to an appropriate amount of the 50 mM Tris solution. All components were mixed by vortexing, and the microreaction tube preheated for at least 5 min in an Eppendorf ThermoStat Plus. Subsequently, an appropriate amount of enzyme stock solution was added to the tube, which was mixed by slight inversions. At given timepoints, a 60 μL sample was removed from the microreaction tube and injected immediately into 940 μL of a 0.2 M NaOH solution in a separate tube to stop the reaction and to dilute the sample simultaneously. After vortexing, 300 μL of the diluted mixture was transferred into UV/Vis transparent 96-well plates. When the concentration of UV/Vis absorbing compound, i.e., deoxythymidine or thymine, was varied, the

sampling volume was adjusted as appropriate to give a constant final concentration of approximately 60 μM UV/Vis absorbing compounds in the alkaline dilutions to generate a UV/Vis absorption in the linear range, i.e., 0–1 absorption units (AU) at 260 nm. The ratio of substrate and product was determined by fitting the spectral 300/277 nm ratio (see below).

UV/Vis absorption spectra were recorded with a PowerWave HT or Synergy MX (BioTek Instruments, Bad Friedrichshall, Germany) in the range of 250–350 nm in 1 nm steps. Spectra were corrected for blanks, i.e., a 0.2 M NaOH solution, recorded within each set of measurements.

### 2.3. Spectroscopic Determination of Deoxythymidine/Thymine Ratio

The deoxythymidine/thymine ratio was determined with a spectrophotometric assay, modified from [21]. In an extension to previous versions of this assay, the spectra were normalized to the isosbestic point of deoxythymidine-thymine mixtures as suggested by [22], which we determined to be at 277 nm. This increased robustness against random dilution errors [23], as they commonly appear in high-throughput experimentation.

Briefly, the spectrum was first blank-corrected by subtracting a spectrum of 0.2 M NaOH, and was subsequently divided by its absorption at the isosbestic point to normalize the spectrum at this position to "1". Then, the normalized absorption at 300 nm was considered as a proxy of the deoxythymidine/thymine ratio.

Thus, the measured absorption ratio $Abs_{300/277} = Abs_{300}/Abs_{277}$ was fitted by a linear relationship without intercept:

$$Abs_{300/277}(\text{experimental}) = x \times Abs_{300/277}(\text{deoxythymidine}) + (1-x) \times Abs_{300/277}(\text{thymine}), \tag{1}$$

where $x$ is the mole fraction of deoxythymidine in the mixture. From pure compound spectra, we determined $Abs_{300/277}(\text{deoxythymidine}) = 0.005115$ and $Abs_{300/277}(\text{thymine}) = 0.772973$.

The algorithms and data treatment functions were implemented in Python 2.7 [24] and Python 3.6 [25]. A snapshot of the software code and the data set used for this work is openly available on zenodo.org and in the Supplementary Material [26–29].

### 2.4. Modelling of the Py-NPase Catalyzed Reaction

The model was implemented as a system of ordinary differential equations in SymPy [30]. The system of equations was wrapped by python-sundials [31] and subsequently integrated by SUNDIALS-CVODE [32]. Parameter estimation was conducted via the lmfit interface [33]. The experimental data handling was performed by in-house Python software, which is equally available from the sources mentioned above.

#### 2.4.1. Cost Functions

In the parameter estimation of the dynamic system (i.e., time courses of the reactions), a weighted-least squares cost function $Z$ was used:

$$Z(k) = \sum_{i=1}^{Q} \frac{1}{\text{Var}(x_i)} \times (c(y_i) - c(x_i))^2 , \tag{2}$$

where $k$ is the parameter set used for calculation of the modelled concentrations; $\text{Var}(x_i)$ is the variance of the $i$-th data point; $c(y_i)$ is the modelled concentration of nucleoside for i-th data point; $c(x_i)$ is the nucleoside concentration as calculated from the experimentally determined mole fraction of deoxythymidine for $i$-th data point, multiplied with $c_{0(x_i)}$, i.e., the designed nucleoside concentration at $t = 0$; and $Q$ is the total number of data points.

For the determination of weights, the 95% confidence interval of data points was set to 5 percentage points of the determined mole fraction as judged by inspection of calibration plots (Figure S1):

$$Var(x_i) = \left( \frac{\sqrt{\varepsilon}}{z_{0.975}} \times x_i \right) \times c_{0(x_i)} , \tag{3}$$

where $\varepsilon = 0.05$ gives the absolute error of the analysis method, and $z_{0.975} = 1.96$ gives the standard score to include 95% of values.

### 2.4.2. Definition of the Differential–Dynamical Model

A schematic visualization of the mechanical model is shown in Figure 1a, with specification into its chemical meaning in Figure 1b. The underlying mechanics of our differential–dynamical model at the process scale can also be represented indirectly by Scheme 1:

$$E + N + P \underset{k_{-1}}{\overset{k_1}{\rightleftharpoons}} EC \underset{k_{-2}}{\overset{k_2}{\rightleftharpoons}} E + S1P + B$$

**Scheme 1. Reaction equation of an enzymatic nucleoside phosphorylation.** Enzyme (E), nucleoside (N), phosphate (P), enzyme complex (EC), pentose-1-phosphate (S1P), free nucleobase (B), reaction rate constants ($k_1$, $k_{-1}$, $k_2$, $k_{-2}$) as defined by Equations (7)–(10).

All steps indicated in the representation of the mechanics are considered elementary step reactions, and, applying law of mass action, the reaction rate equations are derived as the following system of ordinary differential equations:

$$\frac{d[N]}{dt} = \frac{d[P]}{dt} = -r_1 + r_{-1} \tag{4}$$

$$\frac{d[E]}{dt} = -\frac{d[EC]}{dt} = -r_1 + r_{-1} + r_2 - r_{-2} \tag{5}$$

$$\frac{d[S1P]}{dt} = \frac{d[B]}{dt} = +r_2 - r_{-2} \tag{6}$$

where [N] is the concentration of nucleoside (i.e., deoxythymidine), [P] is the concentration of phosphate, [E] is the concentration of free enzyme, [EC] is the concentration of enzyme complex, [S1P] is the concentration of pentose-1-phosphate (i.e., deoxyribose-1-phosphate), and [B] is the concentration of nucleobase (i.e., thymine), with the following rates:

$$r_1 = k_1 \times [E] \times [N] \times [P] \tag{7}$$

$$r_{-1} = k_{-1} \times [EC] \tag{8}$$

$$r_2 = k_2 \times [EC] \tag{9}$$

$$r_{-2} = k_{-2} \times [E] \times [S1P] \times [B] \tag{10}$$

## 3. Results

### 3.1. The Absorption Spectrum of Thymine but Not Deoxythymidine Changes at Alkaline Conditions

The evaluation of enzymatic deoxyribose-1-phosphate forming reactions requires the fast detection of substrates and products. The detection of nucleoside and its corresponding nucleobase by HPLC, and thus the indirect determination of pentose-1-phosphate, has been the standard method to date (e.g., [8,34]). However, it is very time-consuming and laborious and therefore not suitable for use in high-throughput screenings.

We intended to measure the deoxythymidine/thymine ratio by following wavelengths at regions where thymine absorbs at high pH, but deoxythymidine does not, based on an early report [21], and the more recent employment of an UV/Vis assay based on this principle [35]. These are wavelengths >290 nm [36–38]. To correct for varying path lengths which are commonly observed in high-throughput environments based on microtiter plates, and, thus, to make the assay more robust, we normalized the spectra to their isosbestic point, i.e., the point where no change in absorption is observed for any mixture of deoxythymidine and thymine.

To verify this concept experimentally, spectra of pure deoxythymidine, thymine, and mixtures of both were recorded after dilution in NaOH (Figure 2). We then calculated the composition of mixtures from the $Abs_{300/277}$ ratios as described in Materials and Methods. The composition of the full range of mixtures (0–100%, in 10% steps) could be estimated with high accuracy, and the absolute errors between the predicted and actual composition of the mixtures were approximately constant (Figure S1). With this high-throughput tool in hand, we pursued our investigation of a Py-NPase-catalyzed phosphorylation reaction and set out to describe our experimental data in a suitable model.

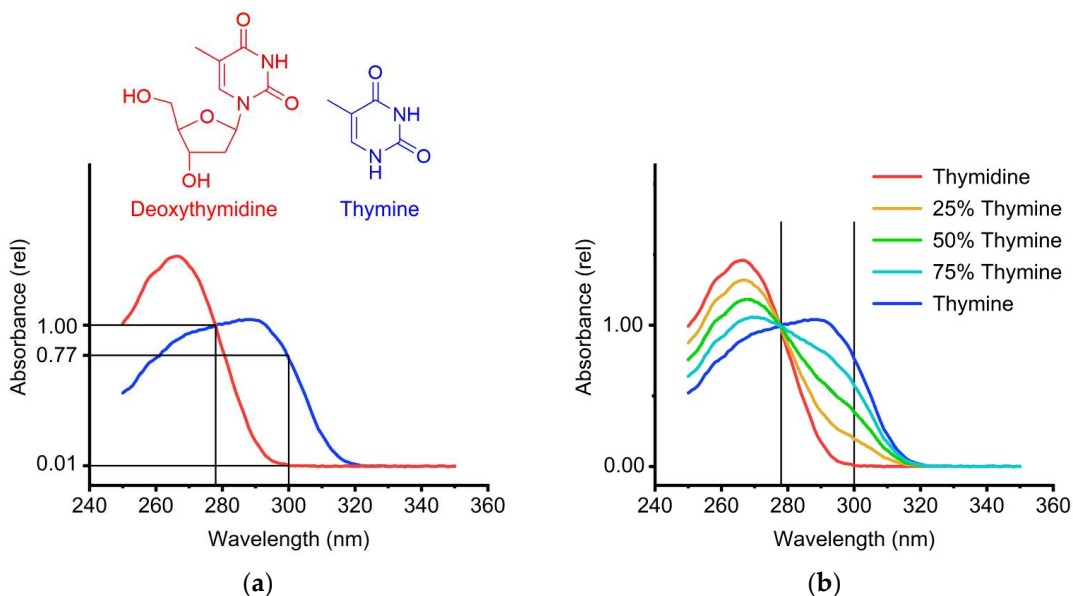

(**a**)  (**b**)

**Figure 2. Comparison of absorption spectra of deoxythymidine and thymine in alkaline dilutions.** Absorption spectra of deoxythymidine and thymine were recorded as described in Materials and Methods in an alkaline dilution at pH 13. The isosbestic point of deoxythymidine/thymine mixtures at 277 nm and the point for determination of the deoxythymidine/thymine ratio at 300 nm are indicated on the x axis. (**a**) The spectra of pure deoxythmidine (red curve) and pure thymine (blue curve) differ significantly when measured in an alkaline dilution. The $Abs_{300/277}$ ratios of pure deoxythymidine ($Abs_{300/277}$(deoxythymidine) ≈ 0.77) and pure thymine ($Abs_{300/277}$(thymine) ≈ 0.01) are indicated on the y axis. The exact values are given in Materials and Methods. Both spectra are shown normalized to the isosbestic point at 277 nm; (**b**) comparison of absorption spectra of pure deoxythmidine, thymine, and indicated mixtures, measured in an alkaline dilution. $Abs_{300/277}$ increases linearly with increasing thymine mole fraction (given as percentage; from red to blue).

### 3.2. Model and Experimental Data Are in Excellent Agreement

Py-NPase-catalyzed phosphorylic cleavage reactions are reversible reactions proceeding towards a dynamic equilibrium. Therefore, the reaction trajectory until equilibrium does not only depend on physical parameters, like temperature and pressure, but also on enzyme concentration, the concentration of substrates, or the presence of products. In order to investigate this enzymatic reaction under biotechnologically relevant conditions, we performed 48 experiments with varying concentrations of enzyme, nucleoside, and phosphate (see Table S1). For our experimental conditions,

i.e., reaction times of 24 h at pH 9.0 and 40 °C, we ensured that the enzyme remained active and deoxyribose-1-phosphate did not degradate (see Figure S2).

To describe the recorded data, a differential–dynamical model was set up. This model allows for the simulation of the concentrations of substrates, products, and enzyme forms over an arbitrarily long time-course. The enzyme reaction can reach a dynamic equilibrium and assumes equal contribution of both substrates to reaction rates and level of equilibrium, as dictated by the underlying law of mass action. We conducted local optimization of the parameters $k = (k_1, k_{-1}, k_2, k_{-2})^T$ (given in unitless numbers for simplicity and in transposed vector form for brevity) to find a parameter set which described the data well. The parameter set we found to perform best on our experimental data is $k = (0.42, 0.17, 0.31, 7.6)^T$. The explicit form ($k_1 = 0.42$ (mM)$^{-2}$ min$^{-1}$; $k_{-1} = 0.17$ min$^{-1}$; $k_2 = 0.31$ min$^{-1}$; $k_{-2} = 7.6$ (mM)$^{-2}$ min$^{-1}$) will be omitted from here on for reasons of brevity.

In the tested range of enzyme and substrate concentrations, we found an excellent agreement between experimental data and our model with this parameter set (Figure 3). The predictions of our models were consistent and evenly distributed around the experimental data points over the whole time course of 24 h (Figures S3 and S4). We could not detect any particular trend of prediction errors towards phosphate, deoxythymidine, or enzyme concentrations. Thus, we conclude that our model is well balanced in the range of experimental conditions described here.

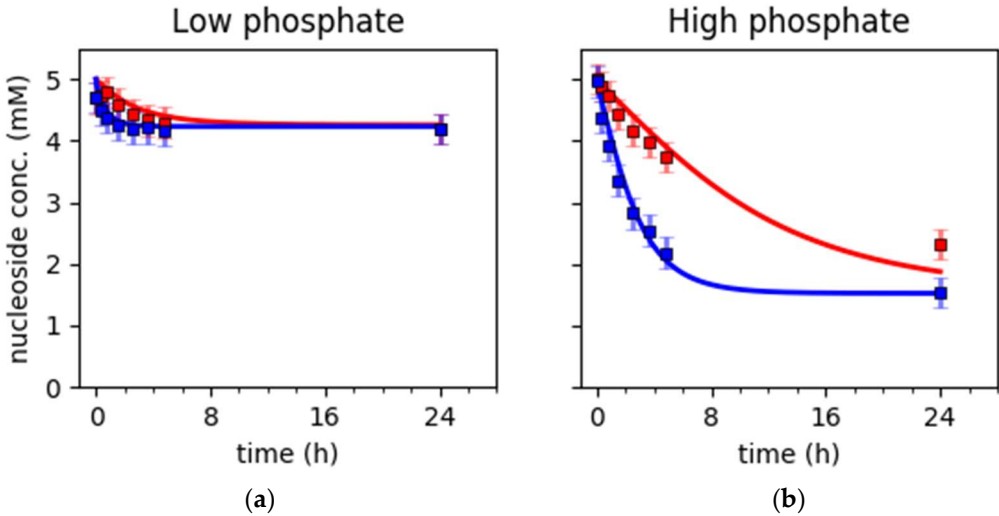

**Figure 3.** **Exemplary fits for experimental data at low and high phosphate concentrations.** (**a**) Experimental data and model predictions for conditions with low phosphate-to-deoxythymidine ratio (2 mM : 5 mM), and varying enzyme concentrations (red: High enzyme concentration, Experiment #13; blue: Low enzyme concentration, Experiment #12). Though the speed of reaction differs in the beginning, both reactions reach the same equilibrium during the time course of the experiment. Error bars represent 95% confidence intervals for the experimentally determined concentrations; (**b**) experimental data and model predictions for conditions with high phosphate-to-deoxythymidine ratio (80 mM : 5 mM), and varying enzyme concentrations (red: High enzyme concentration, Experiment #21; blue: low enzyme concentration, Experiment #20). The two conditions differ in their speed and low enzyme concentration is not sufficient to reach equilibrium. Error bars represent 95% confidence intervals for the experimentally determined concentrations. See Table S1 for experimental condition numbers as given in this figure legend ("Experiment #").

*3.3. Multiple Parameter Sets Can Be Used for the Description of the Phosphorolysis Reaction*

We performed global optimizations with basin-hop and differential evolution algorithms, as well as large-scale local optimizations from widely distributed initial parameter set guesses to find the best global parameter set. We found multiple parameter sets to describe the experimental dataset with almost similar accuracy. Except for $k_2$, which is almost constant, some alternative parameter

sets, e.g., k* = (0.18, 0.12, 0.35, 5.5)$^T$ or k** = (1.4, 0.94, 0.28, 4.6)$^T$, differ drastically from the optimal parameter set k = (0.42, 0.17, 0.31, 7.6)$^T$. However, the cost functions are insignificantly different, with Z(k) = 2.9 × 10$^3$, Z(k*) = 3.0 × 10$^3$, and Z(k**) = 3.2 × 10$^3$ (all values in (mM)$^2$). In practice, this can be attributed to the lacking difference in goodness-of-fit for comparison of simulations for k and the alternative parameter sets, as visualized in Figure 4.

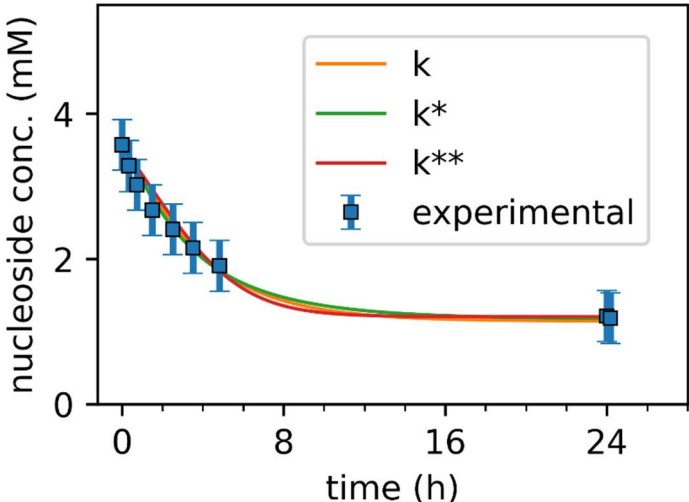

**Figure 4. Non-identifiability of parameter sets from given experimental data.** Experimental data (blue squares) and simulation of parameter sets k, k*, and k**, for a given experimental condition. The modelled results from different parameter sets are almost indiscernible, and, therefore, no decision can be taken on which parameter set is correct.

As the forward reaction is described reasonably well with multiple parameter sets, we cannot decide for any parameter set from our experimental data. This is caused by low sensitivities of the uncertain parameters in regard to our experimental data. As the scope of this work is fulfilled by a model for the forward reaction only, and all parameter sets describe the forward reaction reasonably well, we chose to communicate the parameter set k with lowest value of cost function Z.

*3.4. The Value of the Thermodynamic Equilibrium Constant Is Constant across Methods of Determination*

Finally, we investigated the behavior of the thermodynamic equilibrium constant across all experimental conditions. The thermodynamic equilibrium is approached when there is no observable change in the concentration of the enzyme complex, [EC]:

$$\frac{d[\text{EC}]}{dt} = 0 \, . \tag{11}$$

For our model, this yields two forms to express the equilibrium constant: Either (1) by considering the concentrations of substrates and products at equilibrium:

$$K_{eq} = \frac{B_{eq} \times S1P_{eq}}{N_{eq} \times P_{eq}} \, , \tag{12}$$

or (2) by considering the parameter values:

$$K_{eq} = \frac{k_1 \times k_2}{k_{-1} \times k_{-2}} \, . \tag{13}$$

Estimating the equilibrium constant from the values found in the parameter estimation, one obtains $K_{eq}$ = 0.10. The value of the equilibrium constant is approximately the same for alternative

parameter sets, e.g., k* and k**, emphasizing the principal agreement between multiple parameter sets with the given experimental data.

Similarly, it is possible to derive the equilibrium constant from the equilibrium concentrations of products and substrates, giving a median value of $K_{eq} = 0.10$, similar to the value calculated from kinetic parameters (Figure 5).

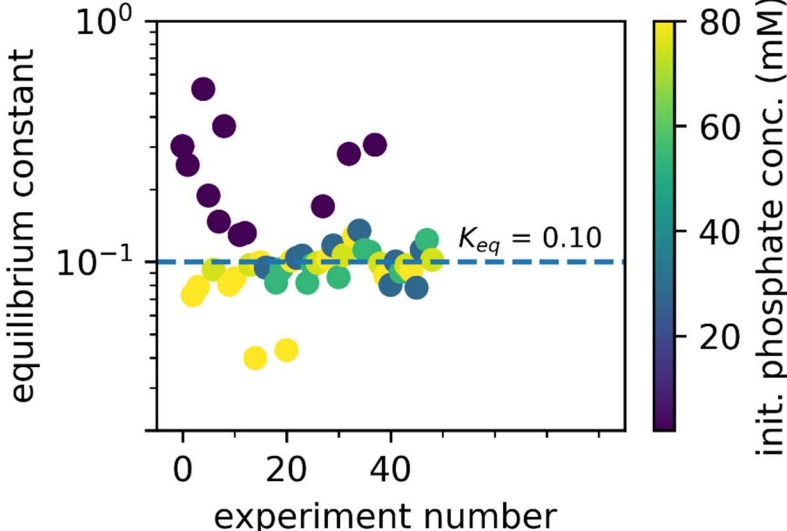

**Figure 5. Equilibrium constant determined at different phosphate concentrations.** Equilibrium constants of the 48 experiments under varying conditions (see Table S1) as determined from the deoxythymidine/thymine ratios of the 24 h data point as described in Materials and Methods. For concentrations of >2mM phosphate, the experimentally determined values from the 24 h data points are evenly distributed around the value calculated from the parameter estimation ($K_{eq} = 0.10$). For experiments with lower phosphate concentrations (dark purple circles), the equilibrium constant is significantly off the calculated value from the parameter sets. The median of all experiments is equal to the value calculated from the parameter estimation. Colored circles: $K_{eq}$ calculated from the 24 h data points, purple to yellow: Increasing initial phosphate concentration; blue dashed line: $K_{eq} = 0.10$, as calculated from the parameter estimation.

## 4. Discussion

To the best of our knowledge, this study presents the first ODE model of an enzymatic two-substrate two-product process. For biotechnological production processes, it is desired to reach equilibrium state conditions to maximize the product yield. For the description of such processes, ODE models are required. In this contribution, we developed such a differential–dynamical model, which places a process perspective onto the enzymatic nucleoside phosphorylation reaction, and which is, regardless of its simplicity, in excellent agreement with our experimental data.

### 4.1. Model Structure

Contrary to Cleland's interpretation of multi-substrate/multi-product enzyme reactions [39–41], which considers multiple enzyme complex intermediates, we modeled the production process as consecutive law of mass actions, and only included one enzyme complex intermediate. Further, we explicitly decided to simplify a probable ordered binding mechanism [10] towards a three-particle collision. In our eyes, these simplifications are justified by the excellent agreement between the experimental data and our model (Figure 3 and Figure S3).

Further elegance of our model is found in its pluggability of equations, which allows for the easy introduction or decommissioning of individual reaction steps. Further, it does not need to rely on

steady-state assumptions, although it is easy to integrate these. Finally, it is easier to provide explicit and precise description of, e.g., inhibitory actions into mechanistic models.

To date, our model does not include terms for the decay of enzyme activity or the degradation of any chemical species. We base these decisions on reports of the exceptional stability in alkaline conditions of deoxyribose-1-phosphate [34] and ribose-1-phosphate [42], as well as on the report of stable enzyme activity over days for thermophilic pyrimidine- [20] and purine-nucleoside phosphorylases [43] at even higher temperatures than those used in this study.

### 4.2. Plausibility of Our Results

To check the correctness of our results, we compared the equilibrium constant (1) from literature and (2) derived from our parameter sets or (3) determined by the equilibrium concentrations of the latest data points. As shown in the Results section, (2) and (3) accord with each other. The values for the equilibrium constant of the Py-NPase catalyzed reaction can be approximated by considering examples from literature [9,10,44,45], being in the similar order of magnitude for related reactions but differing in temperature, buffers, and exact specifications of base and sugar moiety. For a reaction with similar substrates at not too distant experimental conditions, the equilibrium constant was found to be $K_{eq} = 0.102$ at 37 °C and pH 7.4 [7]. This equals the equilibrium constant determined in this work.

Searching for experimental conditions that could discriminate between multiple parameter sets, we found major differences between the parameter sets to be only visible in kinetic study of the backward reaction. Exemplarily, parameter set k** would show significantly faster conversion of deoxyribose-1-phosphate than parameter set k, as $k_{-1}$ is significantly larger. The parameters $k_{-1}$ and $k_2$ can be understood to correlate with the $k_{cat}$ values of the phosphorolysis and synthesis reaction, respectively. Previous work [9] included the progress curve of one phosphorolysis and the corresponding synthesis reaction, and the initial reaction rates can be estimated from the graph given there, being of approximately similar speed. This argument favors k over k**, but for k and k* the situation is less clear. Further research needs to be conducted to resolve this ambiguity.

### 4.3. Limitations and Domain of Validity of the Model

The stability of deoxyribose-1-phosphate and enzyme activity over the assay duration is key to correct conclusions from the experimental data. In addition to reports from literature on pentose-1-phosphate and enzyme stability [20,34,42,43], we performed a control experiment via a coupled read-out, providing evidence for fulfilling these preconditions (Figure S2).

This also clearly points out the domain of validity of the model: In its current form, it applies only in the direction of phosphorolytic cleavage at pH 9.0 and 40 °C for time frames up to 24 h. Outside of this region, one should act on the assumption that corrections will be necessary not only for the temperature- and pH-dependence of the kinetic constants but also in the model structure regarding enzyme inactivation and reactant degradation, especially of deoxyribose-1-phosphate (Figure S3).

### 4.4. Application of Our Results to Production Processes

In the perspective of process control, our model has the potential to describe reactions in a time-resolved fashion, integrating knowledge which was previously not put into equations. The literature is rich in references of successful production processes with nucleoside phosphorylases, but these are typically focused on transglycosylations [35,46–48]. These processes are coupling two nucleoside-phosphorylase reactions, using pentose-1-phosphate as an intermediate in situ; however, for these processes, a prediction of time-resolved process performance was usually not undertaken.

A major advantage of a dynamic model is the ability to optimize processes before or during the run time. Exemplarily, one might want to minimize the amount of consumed enzyme for a batch-process with fixed run time. Our model allows for the calculation of the final yield and required enzyme amount for a fixed run-time, given constraints like, e.g., the solubility of substrate or limiting excess of phosphate (which, for the synthesis of pentose-1-phosphates, is typically used in 1- to 2.5-fold excess

to ease down-stream processing). Similarly, one can calculate the run-time required to reach, e.g., 90% of equilibrium, given an amount of enzyme, substrate, and phosphate. These predictions are the basis of a cost-efficient production.

## 5. Conclusions and Outlook

The determination of the deoxythymidine/thymine ratios with UV/Vis spectroscopy is a fast and cost-effective method for assaying Py-NPase reactions. With this method in hand, we were able to set-up a model capable of describing the time course of Py-NPase reactions for the biotechnological production of deoxyribose-1-phosphate under diverse experimental conditions.

From this, we strive for the predictability of multi-step enzymatic reactions to produce nucleic acid derivatives. Our results pave the way for a significant improvement of production processes towards the synthesis of pharmaceutically interesting nucleosides.

Whenever available, time-resolved information on reaction progress can be used to parametrize the presented model structure. We believe that dynamic modelling will enable efficient process control and reaction engineering, especially when fully parametrized differential–dynamical models for nucleoside phosphorylation reactions are shared within the community.

Especially in multi-enzyme reactions, it will be necessary to integrate terms for undesired reactions, e.g., for product degradation or enzyme inactivation. Our model structure allows for an easy integration of additional terms ("coupling of models"). This would be much less feasible for traditional representations of enzyme kinetics, e.g., in Michaelis–Menten or Cleland notation.

After all, more studies on equilibrium constants and the relationship of kinetic rates at varying experimental conditions, e.g., temperatures or pH values, will be necessary to elucidate the mechanisms of this enzymatically catalyzed reaction further. Dynamic experiments, i.e., varying, for example, the temperature or concentration of reactants, can be next steps for the evaluation and refinement of our results.

**Supplementary Materials:** The following are available online at http://www.mdpi.com/2227-9717/7/6/380/s1 and https://doi.org/10.5281/zenodo.3243519, Figure S1: Further mixtures of deoxythymidine/thymine, and "predicted vs actual" plot, Table S1: Experimental conditions in this study, Figure S2: Degradation progress of deoxyribose-1-phosphate at elevated temperatures, Figure S3: Fits of all experiments, Figure S4: Comparison of inter-day controls.

**Author Contributions:** Conceptualization, R.T.G., M.N.C.B., A.W. and P.N.; Data curation, R.T.G., N.K. and F.K.; Formal analysis, R.T.G. and N.K.; Funding acquisition, R.T.G., M.N.C.B. and P.N.; Investigation, R.T.G., N.K. and F.K.; Methodology, R.T.G., N.K. and M.N.C.B.; Project administration, R.T.G.; Resources, R.T.G., M.N.C.B., A.W. and P.N.; Software, R.T.G. and N.K.; Supervision, R.T.G., M.N.C.B., P.N. and M.G.; Validation, F.K.; Visualization, R.T.G., F.K. and M.G.; Writing—original draft, R.T.G., F.K. and M.G.; Writing—review & editing, R.T.G., N.K., F.K., M.N.C.B., A.W., P.N. and M.G.

**Funding:** RTG was supported by the Berlin International Graduate School for Natural Sciences and Engineering (BIG-NSE) and the Einstein Center for Catalysis ($EC^2$). This research was funded by the Deutsche Forschungsgemeinschaft (DFG, German Research Foundation) under Germany's Excellence Strategy—EXC 2008/1 (UniSysCat)—390540038. We acknowledge support by the German Research Foundation and the Open Access Publication Fund of TU Berlin.

**Acknowledgments:** We thank Athel Cornish-Bowden and María Luz Cárdenas for inspiring discussions and valuable feedback on this manuscript. The authors are grateful to Sarah Kamel (TU Berlin and American University of Cairo) for providing Py-NPase, and to Sebastian Hans (TU Berlin) for sharing computational resources. We thank Mathis Gruber (DexLeChem GmbH) for fruitful discussion in the initial phase of this project.

**Conflicts of Interest:** AW is CEO of the biotech company BioNukleo GmbH. PN is member of the advisory board of BioNukleo GmbH. The funders had no role in the design of the study; in the collection, analyses, or interpretation of data; in the writing of the manuscript, or in the decision to publish the results.

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
