# Peer review of "Dynamic Modelling of Phosphorolytic Cleavage Catalyzed by Pyrimidine-Nucleoside Phosphorylase"

_processes, doi:10.3390/pr7060380_

Round 1
Reviewer 1 Report
The authors describe a very elegant method to determine a kinetic study in conditions close to reality, not using excess of one reagent.
The method is applied here to an enzymatic reaction.
The work is extremely interesting and deserves to be published.
However, to improve a bit the article, some minor suggestions are given here:
- In figure 1:
1a is the model, 1b the studeid reaction. Name the species identically (E,N,P,EC,B, S1P) on 1b (even by leaving the real names) so that non specialists can figure out very quickly the analogy.
- In equations 4 to 10: Write those kinetic equations with the concentrations symbols. Not N but [N], etc.… so that it can be quickly understood by everyone.
- Figure 3 and 4 : Lines should not be so bold.
After this, the article can be accepted.
Author Response
td p { margin-bottom: 0in; }p { margin-bottom: 0.1in; line-height: 120%; }a:link { }
The authors describe a very elegant method to determine a kinetic study in conditions close to reality, not using excess of one reagent. The method is applied here to an enzymatic reaction. However, to improve a bit the article, some minor suggestions are given here: | The authors thank the reviewer for the kind acknowledgement and effort invested into reviewing our manuscript. |
- In figure 1: 1a is the model, 1b the studeid reaction. Name the species identically (E,N,P,EC,B, S1P) on 1b (even by leaving the real names) so that non specialists can figure out very quickly the analogy. | Thank you for the suggestion. We adopted the naming accordingly and hope to have increased the intuitive understandability. |
- In equations 4 to 10: Write those kinetic equations with the concentrations symbols. Not N but [N], etc.… so that it can be quickly understood by everyone. | We adopted the suggested change, and introduced additional clarification at this point of the manuscript. |
- Figure 3 and 4 : Lines should not be so bold. | We changed the strength of the lines according to the reviewer’s suggestion |
Reviewer 2 Report
This paper (Giessman et al.) describes a formulation and experimental verification of a kinetic model (determination of all rate constants) of the reversible phosphorolysis of 2’-deoxythymidine, catalyzed by pyrimidine-nucleoside phosphorylase from thermostable bacteria. The paper is written correctly, and provides new data applicable in the future synthetic works using biocatalysis.
My main objective to this paper is the problem of stability of the product (or co-substrate) of the reaction, deoxyribosyl-1-phosphate, known to be much less stable than the analogous ribosyl-1-phosphate (see ref. [27] from the same lab). The authors should clearly point the limits of applicability of their model (in a separate paragraph or even subsection).
To better estimate parameters of the reverse (synthetic) reaction, the authors could start their assays with non-zero initial concentrations of deoxyribosyl-1-phosphate (synthesized in this lab previously [27]) and free thymine.
Minor points:
1. line 69. The word “unfortunately” should be removed.
2. line 70. Keq for pyrimidine-pyrimidine nucleoside phosphorolytic equilibrium should be provided instead that of hypoxanthine-inosine system.
3. Line 215 (legend to Fig. 2). This legend should clearly point that spectral data refer to (strongly) alkaline conditions. Reference to the experimental section does not seem sufficient.
4. line 102. What was pH of the stock solutions of Thy and dThd?
5. line 110, please explain the meaning of AU.
6. line 154, a reference describing Python is needed.
7. line 201, please remove “of”
8. line 219, “when measured under basic (alkaline) conditions”;
9. lines 237, 239 266, 267 what is the meaning of superscript “T”?
10. line 249, (legend to Fig. 3) “...low and high phosphate...”
11 line 282, 283: do the authors mean steady state or equilibrium?
12. line 341, subscript lacking in “kcat”.
Author Response
This paper (Giessman et al.) describes a formulation and experimental verification of a kinetic model (determination of all rate constants) of the reversible phosphorolysis of 2’-deoxythymidine, catalyzed by pyrimidine-nucleoside phosphorylase from thermostable bacteria. The paper is written correctly, and provides new data applicable in the future synthetic works using biocatalysis. | The authors thank the reviewer for the summary and for the effort invested into reviewing our manuscript. |
My main objective to this paper is the problem of stability of the product (or co-substrate) of the reaction, deoxyribosyl-1-phosphate, known to be much less stable than the analogous ribosyl-1-phosphate (see ref. [27] from the same lab). The authors should clearly point the limits of applicability of their model (in a separate paragraph or even subsection). | The authors agree that deoxyribose-1-phosphate is less stable than some other analogues. In accordance with the results communicated previously where little decomposition of deoxyribose-1-phosphate was observed for 14 days at pH 9.0 and 25°C (ref [27]), we found deoxyribose-1-phosphate to be stable enough at the assay conditions used in this study (pH 9.0, 40°C, 24 hours). This was assayed indirectly (see the new Figure S2) by following conversion over longer times, because direct detection of deoxyribose-1-phosphate is difficult. We found increasing apparent conversion at temperatures of 50 and 60°C (due to decomposition of deoxyribose-1-phosphate), but a very stable equilibrium at 40°C, over time frames vastly exceeding the duration of our assay. We therefore concluded that deoxyribose-1-phosphate was stable during our assay. We added our evidence to the Supplementary Material and introduced an additional explanatory paragraph in the manuscript. |
To better estimate parameters of the reverse (synthetic) reaction, the authors could start their assays with non-zero initial concentrations of deoxyribosyl-1-phosphate (synthesized in this lab previously [27]) and free thymine. | The authors thank the reviewer for the suggestion of future work, with which we fully agree. Studying the “reverse” reaction will be the only way to resolve the ambiguities in parameter sets. Unfortunately, synthesis of deoxyribose-1-phosphate exceeded the time frame granted for the revisions. We also checked commercial suppliers, but delivery times for this substance exceeded this time frame as well. To investigate further possibilities of studying the back reaction, we simulated non-zero initial concentrations of thymine to induce a “back-pressure” of reaction (with simultaneous non-zero initial concentrations of deoxythymidine and phosphate). Although the in silico analysis showed information to be gained through this, it would have not been enough information to discriminate between the parameter sets. We would like to address the reviewer’s suggestions in our future research, and would like to share the available data with the scientific community in its current form. |
Minor points: | |
1. line 69. The word “unfortunately” should be removed. | This suggestion was incorporated into the manuscript. |
2. line 70. Keq for pyrimidine-pyrimidine nucleoside phosphorolytic equilibrium should be provided instead that of hypoxanthine-inosine system. | We are very grateful for the reviewer’s suggestion, as additional literature review yielded indeed a (well hidden) value for the exact reaction under investigation in this manuscript. We further generalized the discussed statement by giving a range of equilibrium constants from multiple sources, which embrace a multitude of reactions, to showcase the general problem of phosphorolytic cleavage of nucleosides. |
3. Line 215 (legend to Fig. 2). This legend should clearly point that spectral data refer to (strongly) alkaline conditions. Reference to the experimental section does not seem sufficient. | This suggestion was incorporated into the manuscript. |
4. line 102. What was pH of the stock solutions of Thy and dThd? | This suggestion was incorporated into the manuscript. |
5. line 110, please explain the meaning of AU. | This suggestion was incorporated into the manuscript. |
6. line 154, a reference describing Python is needed. | This suggestion was incorporated into the manuscript. |
7. line 201, please remove “of” | This suggestion was incorporated into the manuscript. |
8. line 219, “when measured under basic (alkaline) conditions”; | This suggestion was incorporated into the manuscript. |
9. lines 237, 239 266, 267 what is the meaning of superscript “T”? | This suggestion was incorporated into the manuscript. |
10. line 249, (legend to Fig. 3) “...low and high phosphate...” | This suggestion was incorporated into the manuscript. |
11 line 282, 283: do the authors mean steady state or equilibrium? | This suggestion was incorporated into the manuscript. |
12. line 341, subscript lacking in “kcat”. | This suggestion was incorporated into the manuscript. |